# Assessment of Cold Atmospheric Pressure Plasma (CAPP) Treatment for Degradation of Antibiotic Residues in Water

**DOI:** 10.3390/antibiotics12071115

**Published:** 2023-06-28

**Authors:** Ewa Wielogorska, Padrig B. Flynn, Julie Meneely, Thomas P. Thompson, William G. Graham, Brendan F. Gilmore, Christopher T. Elliott

**Affiliations:** 1Institute for Global Food Security, Queen’s University Belfast, Belfast BT9 5DL, UK; j.p.meneely@qub.ac.uk; 2School of Pharmacy, Queen’s University Belfast, Belfast BT9 7BL, UK; p.flynn@qub.ac.uk (P.B.F.); t.thompson@qub.ac.uk (T.P.T.); 3Centre for Plasma Physics, School of Mathematics and Physics, Queen’s University Belfast, Belfast BT7 1NN, UK; b.graham@qub.ac.uk

**Keywords:** cold atmospheric pressure plasma, surface water, antibiotics, antibiotic resistance

## Abstract

The presence of antibiotic residues in water is linked to the emergence of antibiotic resistance globally and necessitates novel decontamination strategies to minimize antibiotic residue exposure in both the environment and food. A holistic assessment of cold atmospheric pressure plasma technology (CAPP) for β-lactam antibiotic residue removal is described in this study. CAPP operating parameters including plasma jet voltage, gas composition and treatment time were optimized, with highest β-lactam degradation efficiencies obtained for a helium jet operated at 6 kV. Main by-products detected indicate pH-driven peroxidation as a main mechanism of CAPP-induced decomposition of β-lactams. No in vitro hepatocytotoxicity was observed in HepG2 cells following exposure to treated samples, and *E. coli* exposed to CAPP-degraded β-lactams did not exhibit resistance development. In surface water, over 50% decrease in antibiotic levels was achieved after only 5 min of treatment. However, high dependence of treatment efficiency on residue concentration, pH and presence of polar macromolecules was observed.

## 1. Introduction

Climate change and increasing human global population, expected to reach 9.6 billion by 2030, contribute to significant concerns regarding both food safety and security [1]. Thus, increased efforts are directed towards improving agricultural productivity and more efficient management of current resources [2,3] while assuring consumers’ safety via continuous monitoring of a variety of man-made and natural chemical residues in food commodities and the environment itself [4,5]. In the past four decades, there has been an increase in the use of a number of pharmaceuticals in animal husbandry to improve growth performance, including anabolic steroids and antimicrobials [6]. The excessive use of antibiotics in clinical, agricultural and veterinary settings has led to the emergence and dissemination of antimicrobial and antibiotic-resistant bacteria. Accordingly, the incidence of antibiotic-resistant infections are rising annually and current projections estimate as many as 10 million deaths annually by 2050 with substantial associated costs of treating such infections of USD 20 billion per year in the US alone [7]. Recent analysis of the global burden of bacterial antimicrobial resistance (AMR) predicted that 1.27 million deaths were attributable to bacterial AMR in 2019 [8]. The World Health Organization (WHO) have highlighted AMR as one of the biggest threats to global health, food security and development [9]. Antibiotic-resistance hotspots were identified in medical settings but also in the environment, where exposure to anthropogenic use of antibiotics is a feature, including agricultural and aquaculture facilities, municipal and wastewater systems and pharmaceutical manufacturing effluents [10]. The release of active pharmaceutical ingredients (APIs), including antibiotics, into the natural environment occurs during manufacture, use and disposal (including in sewage and animal waste), leading to disruption of natural ecosystems, and is considered a global threat to human and environmental health [11,12], with low- to middle-income countries worst affected by API pollution. Antibiotics are ubiquitous contaminants and contribute to antibiotic-resistance gene (ARG) reservoirs in the environment [13], with antibiotic concentrations in many rivers contaminated by sewage thought to be sufficient to drive the emergence of ARGs [12]. Broad-spectrum penicillins and cephalosporins make up 55% of total antibiotic consumption globally [14]. Consequently, a large proportion of EU fresh waterways have been reported to be contaminated with sub-µg/L concentrations of antibiotics, with β-lactams being one of the most frequently detected residues [15] resulting in detection of resistance towards β-lactam antibiotics detected in high frequencies in treated wastewater microbial populations [16].

The European Union enforces stringent regulations on the presence of pharmaceutical residues in animal-derived food, where samples are considered non-compliant if they contain 1% or more of such residues [17,18]. However, there have been recent reports of samples that not only exceed this limit but contain considerably high percentages of antibiotic residues [19,20,21]. These findings are particularly concerning as such residues could potentially contribute to antimicrobial resistance (AMR) [22]. To limit environmental exposure to antibiotic residues in hospital and industrial wastewater effluents, and limit food wastage through rejection of non-compliant produce, novel technologies enabling fast and efficient detoxification are being currently explored. A number of approaches have been assessed in the literature including photolysis, peroxidation, ozonation or chlorination [23,24,25,26]. Nevertheless, municipal wastewater-treatment systems are not designed to and are unable to fully remove antibiotic (and other API) residues [27]; modifications of conventional wastewater treatment are still being explored including membrane bioreactors [28] or hybrid carbon membranes [29]. Plasma is classified as a unique state of matter, of wholly or partially ionized gases, comprised of various reactive atomic and molecular species, including neutral atoms and molecules, ions, radicals, excited state electrons and photons [30]. The ability to generate plasmas at or near room temperature, and production of such a diverse mixture of reactive species, has led to nascent fields of research in plasma chemistry, plasma medicine, material engineering, as well as emerging environmental and agricultural applications [31], including wastewater decontamination from surfactants, personal care products, pharmaceuticals, dyes or disinfectants [32,33,34]. Nevertheless, few studies have focused on β-lactam antibiotics, with a limited number of compounds assessed [35,36,37]. The present study evaluates the efficiency of cold atmospheric pressure plasma (CAPP) technology in the degradation/removal of a panel of β-lactam antibiotic residues in both surface water and milk, contributing to the efforts to tackle antibiotic resistance and food safety and security issues through modulation of aqueous antibiotic residues. A complex assessment of residue degradation, by-product formation, their toxicological characteristics and ability to induce resistance in *E. coli*, alongside examination of the influence of matrix constituents on degradation efficiency, are described, employing high- and low-resolution mass spectrometry as well as in vitro bioassays.

## 2. Results and Discussion

### 2.1. Decontamination Mechanism and Efficiency in Solvent

In most cases, the decomposition of both penicillins and cephalosporins follows first-order kinetics. However, at higher concentrations the degradation kinetics of certain compounds shift towards a zero-order reaction (Figure 1), especially visible for amoxicillin. The gradual accumulation of reactive species does not seem to influence the decomposition rate; hence, the fitted equations explain the dynamics in a worse manner, yielding higher prediction errors. Associated differences in treatment efficiency can vary from 2% to up to 51%, resulting from 10-fold change in residue concentration for dicloxacillin and amoxicillin, respectively, after 10 min of He jet exposure (Figure 1).

This variation may be accounted for by increasing the system’s operating voltage/gas flow; however, those results underline the need for careful assessment of decontamination efficiency at different concentrations, even within the same family of compounds as the susceptibility to the treatment may vary. Since the most reliable curve fitting was obtained for 25 nM, with coefficients of determination ranging from 0.96 to 1.00, those reaction rates were employed in the comparison of treatments (Figure 2).

The fastest degradation was achieved for He jet operated at 6 kV with reaction rates ranging from 0.0121/s to 0.0061/s, translating to half lives between 0.95 min and 1.93 min, with most and least sensitive antibiotics being cefquinome and amoxicillin, respectively. A clear pattern of treatment efficiency in water can be noted for all the compounds with mixed oxygen jet, and helium jet with voltage of 5 kV having lower efficiency than helium jet operating at 6 kV (both *p* < 0.001) with respective reaction rates lower by 47% and 38% on average. In agreement with previous reports, lower reactive species concentrations and so decomposition efficiencies can be expected at lower operating voltages and oxygen presence in the gas mixture [38]. Statistically, there is no difference between He, 5 kV jet and He/O_2_, 6 kV jet decomposition rates, which indicates that introduction of only 0.5% of oxygen into feed gas has a similar effect, that is, a decrease in applied potential difference of 1 kV. In the mixed gas jets, oxygen concentration is associated with the decrease in the total current and so plasma density, resulting in a decrease in OH radical, NO and Hα concentrations at the expense of an increase in ozone and O radical formation due to higher electronegativity of oxygen [39]. A similar response was noted and described in the system employed herein [40].

During our experiments, we observed a significant influence in the buffering capacity and pH on the decomposition rates. A notable pH drop was witnessed during the plasma treatment in water, from 7 to 4.5 over the course of a 10 min treatment. Conversely, the pH in PBS remained constant at 7.5 throughout the treatment. The differential pH conditions between water and PBS influenced the reaction rates and decontamination efficiency, with a marked decrease in rates by 56% on average. This effect was more pronounced for cephalosporins (65 ± 8%) than penicillins (46 ± 11%) [41,42]. One of the main reactive species present in water during plasma treatment was shown to be hydrogen peroxide and an increase in peroxidation efficiency was previously correlated with pH decrease [43]. We attribute the pH-dependent efficiency to enhanced peroxidation at lower pH values, which is considered the main mechanism for the degradation of β-lactam antibiotics in our setup. Hence, the influence of pH and buffering capacity plays a crucial role in the plasma treatment’s overall effectiveness and needs to be considered when designing and interpreting such decontamination processes.

Consequently, additional modeling was performed to correlate energies on the highest occupied molecular orbital (HOMO) with obtained reaction rates in water. Performed regression (Figure 3) revealed moderate (R^2^ = 0.76) dependence on the HOMO energies, similar to previous report on β-lactams oxidation by peracetic acid [19], offering some indication of oxidation potential within the β-lactam compounds group, which could be employed as a predictor of treatment efficiency for other compounds within the group. 

### 2.2. By-Product Formation

Two model compounds were picked for by-product-formation assessment from each group, i.e., amoxicillin (AMX) and cloxacillin (CLX) from the penicillin group and cefoperazone (CFPZ) and cefazolin (PFZ) from the cephalosporins group. The concentrations required to observe the by-products via UPLC-HRMS were higher (around 100 times) than tested during kinetic studies, due to lower sensitivity of HRMS at hand, with solutions exposed for 5 min. Obtained chromatograms are presented in Figure 4.

Due to the presence of sulphur on both penam and cepham rings as well as two additional sulphur atoms at the thiadiazole ring terminus in the case of cephalosporins, those reaction sites will be most prone to electrophilic attack. A clear pattern can be observed in both groups treated, i.e., at least two peaks with a mass difference of +16 m/z (365 m/z at 4.13, 1.74 and 4.78 min, for AMX and 452 m/z at 11.27 min and 12.33 min for CLX, 471 m/z at 6.41 and 6.53 min for CFZ and 662 m/z at 7.97 and 8.13 min for CFPZ) or +32 m/z (381 m/z 3.13 min for AMX), which indicates isomeric compound formation resulting from oxygen addition. In the case of penicillins, the highest electron densities can be noted on the penam ring; as such, we postulate that it is the main site of sulfoxidation, with similar findings reported for plasma-treated ampicillin [36]. Additional minor isomers detected may be the products of hydroxylation of the terminal phenol ring due to hydroxyl radical attack. In the case of cephalosporins, higher electron densities were noted on the thiadiazole ring and neighboring sulphur; we postulate that those are the sites of sulfoxidation in this group. Nevertheless, other minor by-products can be noted for CLX (438 m/z at 9.36 min and 454 m/z at 9.62 min) and CFZ (211 m/z at 7.89 min). For CLX, we postulate that those by-products are a result of hydrogenation and penam hydrolysis, while the 211 m/z CFZ by-product must be a result of more pronounced molecule decomposition. However, those are minor products of CAPP treatment, which under studied conditions generally does not cause β-lactam ring opening. However, it introduces some modification which should alter molecules’ binding to penicillin-binding proteins, in essence decreasing residues’ contribution to antibiotic resistance [44].

### 2.3. Toxicological and Antibiotic-Resistance Assessment

In terms of AMX toxicity, no significant results were obtained during HCA (Figure 5), with the exception of a slightly heightened nuclear area (NA) noted for untreated AMX solution, differing by 16% when compared with solvent control (*p* < 0.05). Even though the concentration tested was similar to those detected after oral administration of the antibiotic, some cellular stress was detected. However, this low response in the assay did not allow one to come to a conclusion concerning whether AMX solution exposure to CAPP resulted in a decrease in toxicity as the obtained result was not significantly different (*p* < 0.05) from either the solvent control or the AMX untreated solution. Nevertheless, no increased toxicity was noted for the treated AMX solution. None of the milk extract expressed any toxicity, suggesting that CAPP treatment does not result in toxic by-product formation in milk matrix.

To confirm that the CAPP-mediated modifications of AMX do not contribute to antibiotic resistance, evaluation of its antimicrobial activity and an evolution-resistance study were performed using the Gram-negative organism—*E. coli.* The generation of antibiotic-resistant bacterial populations from exposure to sub-MIC concentrations is considered a greater threat and a more challenging task to manage than traditional selection of resistance at high MIC concentrations. This is in part due to these resistant sub-populations efficiently managing the fitness cost of resistance and maintaining resistance even without the presence of antibiotics. CAPP treatment of a 256 µg/mL AMX solution for 20 min in water resulted in 80% decomposition of AMX, yielding the most efficient formation of detected by-products (Table 1, Figure 4a). It was therefore chosen for the assessment of by-product contribution to the evolution of resistance to AMX at sub-MIC values. Figure 6 depicts the results from the evolution-resistance study; for each lineage, a two-fold increase in the MIC occurred for *E. coli* continually passed into sub-MIC concentrations of AMX (0.5 µg/mL). This increase occurred at different times for each lineage, reflecting the biological variation of bacteria even though they arose from the same strain and colonies. There was no observed increase in the MIC value for CAPP-treated AMX, indicating that the formation of CAPP-mediated AMX by-products did not put the bacteria under selective pressure for the development of AMX resistance. In addition, it would also suggest that CAPP treatment of AMX resulted in a reduction in AMX that was below the minimum selective concentration, that is, the concentration of AMX that results in a benefit in growth with respect to carrying the resistant trait over the fitness cost of carrying the same trait.

### 2.4. Plasma Efficiency in Residue Removal in the Matrix

The efficiency of plasma application in antibiotic residue removal was assessed both in surface water and milk. Concentrations that spiked in surface water oscillated around 10 ng/g (0.025 µM) and were around 150 times higher than those reported in surface waters [27] but similar to those reported in raw wastewater-treatment plant influent [28]. Higher concentrations were employed due to sensitivity limits of the LC-MS/MS method with preferred fast and simplified sample-preparation procedures due to ongoing reactive species activity in the exposed samples.

Milk was assessed at the level three times higher than the legislated MRLs, oscillating between 12 and 375 ng/g (Appendix A), to mimic concentrations in non-compliant samples. Percentages of residues remaining after 5 min of plasma exposure are presented in Figure 7. In surface water, levels of all residues were decreased on average by 53%, with a pattern resembling results achieved in PBS. Very limited antibiotic residue removal can be noticed in milk, with a significant (*p* < 0.05) decrease noted only for oxacillin, cloxacillin, dicoxacillin, cephalexin, cefaperazone and cefacetrile, ranging from 2% to 8%.

In terms of milk matrix small molecular mass constituents, presented principal component analysis (PCA) (Figure 8) shows clear discrimination between both spiked and treated groups versus their respective controls. The created model explains almost all variation detected in the tested samples (R2(PC1-3) = 94%); however, it is based on features that are most significantly different (power 0.9995), in this case accounting for only 10% of all detected features (n = 1174). As mentioned previously, the composition of the matrix has a great influence on the removal efficiency with even low protein content dramatically reducing CAPP treatment effectiveness due to reactive species attenuation by polar macromolecules. Similarly, any buffering capacity of the medium exposed will also slow down residue peroxidation at a non-acidic pH. As such, CAPP application in any liquid system will require substantial modeling and real-time monitoring to account for changes in treated solution composition and modify CAPP settings to sustain required removal efficiency.

## 3. Materials and Methods

The following were purchased from Sigma-Aldrich: amoxicillin trihydrate VETRANAL™ (purity 98.3%), cephalexin monohydrate (purity 93.5%), amplicillin trihydrate European Pharmacopoeia (EP) reference standard (purity 99.5%), cefazolin EP reference standard (purity 99.6%), cefquinome sulfate VETRANAL™ (purity 99.1%), cefapirin sodium salt (purity 100%), cefoperazone dihydrate EP reference standard (purity 94.2%), ceftiofur hydrochloride VETRANAL™ (purity 99.3%), cefacetrile VETRANAL™ (purity 98.6%), penicillin G potassium salt VETRANAL™ (purity 99%), oxacillin sodium salt monohydrate VETRANAL™ (purity 99.1%), cloxacillin sodium salt monohydrate VETRANAL™ (purity 96.0%), nafcillin sodium salt VETRANAL™ (purity 97.4%), dicloxacillin sodium salt monohydrate VETRANAL™ (purity 99.0%), penicillin G-d7 N-ethylpiperidinium salt VETRANAL™ (purity 99.6%). The rest were acquired from Toronto Research Chemicals: desfuroyl cephapirin (DAC) sodium salt (purity 95%), desfuroyl deftiofur dysteine disulphide (DCCD), cefalonium hydrate (purity 95%), amoxicillin-d4 (major) (purity 96%), cephalexin-d5 hydrate (purity 98%), nafcillin-d5 sodium salt (purity 95%). Organic solvents such as LC-MS Chromasolv^®^ methanol (MeOH) and acetonitrile (MeCN), HPLC grade dimethyl sulfoxide (DMSO), hexane as well as eluent additives for LC-MS, formic and acetic acid, 25% ammonium hydroxide were purchased from Sigma-Aldrich (St. Louis, MO, USA). Ultra-pure water 18.2 MΩ/cm was generated in house employing a Milli-Q^®^ system (Merck Millipore, Billerica, MA, USA). Two UPLC columns were assessed during LC method development, i.e., Luna Omega-Polar 1.7 μm, 100A, 2.1 mm × 100 mm from Phenomenex (Macclesfield, UK) and Cortecs^®^ T3 1.6 μm, 2.1 mm × 100 mm from Waters (Manchester, UK). Amber and silanised LC-MS vials were purchased from Waters (Manchester, UK). Other laboratory equipment employed included DV215CD analytical balance (Ohaus Europe GmbH, Nanikon, Switzerland), DVX-2500 multi-tube vortexer (VWR International, Lutterworth, UK), MIKRO 200R centrifuge (Hettich UK, Salford, UK), TurboVap LV and SPE Dry 96 well plate evaporators (both from Biotage, Uppsala, Sweden).

### 3.1. Milk and Surface Water Samples

Grab surface water samples were obtained from the Lagan River, around 1.5 km down-stream the Newtownbreada wastewater-treatment plant. The samples were stored at −20 °C for four days before the analysis. For the blank milk samples, a full fat commercial milk was employed, with non-detectable analyte levels. The LC-MS/MS method was validated employing 20 blank, raw milk samples collected from different cows, aliquoted and stored at −80 °C until the time of the analysis.

### 3.2. Sample Preparation

Spiked deionized water and surface water (after centrifugation at 10,000 g for 5 min) samples were injected directly onto the LC-MS/MS system. Milk samples were thawed and 0.5 g was transferred into 2 mL Eppendorf tubes. Then, differing volumes of working standard solution, ranging from 100 μL to 10 μL, were added to the blank sample aliquots to create a six-point calibration curve (including zero). All samples were then spiked with 50 μL of internal standard solution. Samples were allowed to stand for 10 min and then were extracted with 1.5 mL of MeCN via 10 min of vortexing at 2500 rpm on a mulit-tube vortexer. Next, samples were centrifuged at 10,000× *g* at 4 °C and 1 mL of the aliquot was transferred to a fresh 1.5 mL Eppendorf tube. The solution was then dried down under a gentle stream of nitrogen to 250 μL. To remove the lipids, 250 µL of hexane was added to the aliquots and vortexed at 2500 rpm for 2 min, then centrifuged for 2 min at 10,000× *g* at 4 °C to facilitate phase separation. Then, the solvent was removed and 200 μL of water layer was transferred to a silinized LC-MS vial. The samples were placed under nitrogen for 10 min to remove residual hexane and then diluted with 250 μL of water, before injecting onto the system.

For the untargeted analysis, milk samples were processed in the same manner but without the hexane wash and further sample dilution with water. To ensure repeatability, both milk extracts’ and solvent samples’ mass was equalized on the analytical balance to ±1% both after plasma treatment, to correct for water evaporation.

### 3.3. Atmospheric Pressure Plasma System

The CAPP source employed herein was developed in house in the Centre for Plasma Physics, School of Mathematics and Physics, Queen’s University Belfast, UK, and based on the design of Teschke et al. [45] and previously described and characterized in detail [39,40]. Briefly, a dielectric barrier discharge (DBD) jet was created in a quartz tube (inner diameter 4 mm and outer diameter 6 mm), mounted vertically, with a helium/(0.5%) oxygen gas mixture flowing through the tube at 2 standard liters per minute (SLM). Two copper electrodes (2 mm wide) encircle the tube, separated by 25 mm. The up-stream electrode is grounded. When the voltage, pulsed at 20 kHz and with an amplitude of 5 or 6 kV, is applied to the down-stream electrode, a plasma is ignited within the quartz tube and the effluent (afterglow region) exits the tube in the direction of the gas flow.

### 3.4. Plasma Exposure of Solvent Solutions, Surface Water and Milk

Water solutions of analytes were prepared on the day of the analysis by spiking intermediate standard solutions to obtain final analytes’ concentrations of 250, 125 and 25 nM. In a 1.5 mL Eppendorf tube, 1 mL aliquots were exposed to the plasma jet for 1, 5 and 10 min at a distance of 15 mm from the surface of the solution to the jet outlet. Surface water was spiked at 25 nM, while milk samples were spiked at three times maximum residue level (MRL), (Appendix A) and exposed in a similar matter. The exposures were performed in triplicate on three different days. To characterize the degradation kinetics of analytes, acquired data sets were fitted with an exponential function (first rate kinetics) to obtain degradation rates and antibiotics half lives under plasma treatment employing GraphPad Prism software; the graphs were constructed employing normalized data to better visualize differences in treatments, while degradation rates were calculated based on measured molar concentrations employing the following equation:*C*_t_ = *C*_0_∙*e*^−*kt*^(1)
where *C*_t_ is the residual concentration of the toxin at a given time point, *C*_0_ is the initial concentration of the toxin, *t* is time and *k* is the reaction rate.

For the decomposition by-product elucidations via HRMS, separate solutions of each toxin were prepared from the stock solution aliquots at 10 µg/mL on the day of the analysis. Solutions were exposed to the plasma jet in the same manner at two time points of 5 and 10 min.

All samples were stored on ice after the exposures to slow down residual activity and analyzed within 2 h of the plasma exposure.

### 3.5. UPLC-MS Instrumental Set-Up

Analyte separation was performed on a Waters Acquity UPLC I-Class system (Milford, MA, USA), employing a Luna Omega-Polar (1.7 μm, 100Å, 2.1 mm × 100 mm) from Phenomenex (Macclesfield, UK) maintained at 40 °C. The pump was operated at a flow rate of 0.4 mL/min and mobile phases consisted of (A) 0.2 mM ammonium acetate with 0.01% HCOOH in H_2_O (*v*/*v*) and (B) 0.01% HCOON in MeCN, due to previously reported sensitivity enhancement [35]. For the separation of the analytes, the following gradient was applied: isocratic 0–1 min 99% A, linear 1.0–3.0 min 75% A, 3.0–6.5 min 40% A, column flush 6.5–7.5 min 1% A, held for 1 min, 8.5–9.5 min 99% A held for 2.5 min for column equilibration with total run time of 12 min. The injection volume was 10 μL with needle placement set to 0.3 mm from the bottom of the vial. The UPLC autosampler temperature was maintained at 6 °C, while the needle was purged and washed after pre- and post-injection with mobile phase A and 0.1% HCOOH in H_2_O/MeOH/MeCN (2:1:1) solution, respectively.

Spectrometric analysis was performed on a Waters Xevo TQ-S triple quadrupole mass analyzer (Manchester, UK) equipped with an ESI ionization probe, operating in positive ionization mode. Manual optimization of spectrometric conditions was performed via direct infusion of standard solutions (at concentration of 1 μg/mL).

The following settings were applied: capillary voltage was set at 2.5 kV, the desolvation and source temperatures were set at 550 and 130 °C, respectively, while nitrogen cone and desolvation flow rates were set to 320 and 1000 L/hr. Argon was employed as a collision gas, with a flow of 0.24 mL/min, yielding a collision cell pressure of 2.5 × 10^−3^ mBar. Inter-scan and -channel delays were both set to 3 ms, while dwell times ranged from 30 to 200 ms. Inter-scan delay and inter-channel delay were set to 0.003 s.

The cone voltage and collision energy were optimized for each analyte and a minimum of two product ions were selected so that a minimum of four identification points were obtained for all the analytes, as required by Commission Decision 2002/657/EC.

In order to investigate antibiotics-decomposition by-products as well as plasma-treatment influence on the milk matrix components, both antibiotics solution in solvent and milk extracts exposed to plasma were analyzed, employing high-resolution mass spectrometry. The chromatographic separation was performed on Waters Acquity I-Class UPLC (Milford, MA, USA) equipped with an Acquity Cortecs T3 column (100 mm × 2.1 mm, 2.7 μm), maintained at 45 °C. The pump was operated at a flow rate of 0.4 mL/min with mobile phases (A) 0.1% formic acid in water and (B) 0.1% formic acid in methanol with injection volume at 5 μL. A linear gradient was set as follows: 1.50 min of 99% (A) to 99% (B) over 18 min with final 2 min at initial conditions for column equilibration. Mass spectral data were attained using Waters Xevo G2-XS QTof mass spectrometer (Manchester, UK) with an electrospray ionization source operating in positive mode over the range 50–1200 m/z with a lock–spray interface for real-time accurate dual-point mass correction with Lecuine-Enkephalin.

The UPLC-MS/MS and HRMS systems were controlled via MassLynx software (V4.1) with targeted method results processed with TargetLynx software (V4.1), while untargeted data were extracted, deconvoluted and normalized with Progenesis QI, including subsequent statistical analysis employing UV-scaled data analyzed via principal component analysis (PCA).

### 3.6. Preparation of Standard Solutions and Quality Control Samples

The majority of stock solutions were prepared by dissolving the appropriate amount of standard antibiotic powder in either mixtures of H_2_O/MeCN or DMSO as described [46], taking into consideration the purity of each particular batch of material, to obtain a concentration of 1 mg/mL with the exception of amoxicillin and DCCD which were prepared at 0.5 mg/mL, while DAC was prepared at 2 mg/mL. Deuterium-labeled standards were dissolved in D2O/MeCN-d4 to avoid isotope exchange. Each standard solution was divided into aliquots and stored in amber vials at −80 °C for single use only. Fresh stocks were prepared every 3 months due to reported low stability of β-lactam antibiotics [42,47]. Working standards were prepared and stored in a similar matter.

Standard analysis set-up for UPLC-MS/MS milk analysis included an extracted blank sample, a zero sample—a blank matrix spiked with internal standard mixture only—a set of five extracted matrix-matched calibrants extracted on the day of the analysis (as described in Section 2.2.) and two recovery controls, spiked post-extraction at calibration levels 2 and 4 (CL2 and CL4). Additionally, blank solvent as well as the solvent equivalent of CL2 were injected every five samples to monitor the performance of the equipment during the run. For the analysis of solvent solutions, a calibration curve with a minimum of five points was injected on the day of the analysis to ensure the linearity within 100% to 1% of the tested concentration(s).

For the untargeted UPLC-HRMS analysis, a pooled milk extract sample was injected every five samples to monitor for possible instrumental drifts with randomized sample order.

### 3.7. LC-MS/MS Method Validation

The method was validated according to Commission Decision 2002/657/EC and performed studies included selectivity, linearity, matrix effect, accuracy, reproducibility (WRL), decision limit (CCα) and detection capability (CCβ)

The selectivity of the method was investigated through injecting standard solutions of all analytes and internal standards individually and through testing 20 milk samples from different animals in order to check the presence of any interferences eluting at and around the retention times of the analytes. The linearity of the curves was considered satisfactory if R2 ≥ 0.99 and if individual residuals did not deviate by more than ±15% from the calibration curve with the lowest point not deviating more than ±20%. The accuracy of the analytical method was assessed via analysis of 20 different blank samples, which for the MRL substances were assessed at 0.5, 1 and 1.5 times the MRLs established by current legislation, except for DAC and DCCD, which were validated at 60 and 50 ng/g, respectively. The WLR study was performed on three separate days with 18 different negative samples randomized before fortification at each validation level. The available labeled compounds were used in the quantification of their corresponding analytes. Additionally, cloxacillin, dicloxacillin and oxacillin were all corrected using nafcillin-D5; cefacetrile was corrected using cefazolin-13C215N; cefalonium was corrected using cephalexin-D5. No internal standards were used for the other analytes, as the available labeled compounds were found to be unsuitable in their quantification. For the MRL substances, CCα and CCβ were calculated from the within-laboratory reproducibility data, while matrix effects were evaluated by spiking 20 different blank samples post-extraction at the MRL level and the signal obtained from those samples was compared to the signal obtained from a standard solution at the same concentration.

### 3.8. High Content Analysis

To assess the cytotoxicity of amoxicillin as well as extracts, both spiked and blank maize exposed to the CAPP were tested on human hepatocarcinoma (HepG2) cells employing High Content Analysis (HCA). Multiple parameters including cell number (CN), nuclear area (NA), nuclear intensity (NI), mitochondrial mass (MM) and mitochondrial membrane potential (MMP) that are indicators of cell health were simultaneously evaluated. The cells were maintained in media in 75 cm^2^ cell culture flasks (Nunc, Roskilde, Denmark) at 37 °C in a 5% carbon dioxide atmosphere (*v*/*v*). Cells were cultured in DMEM media supplemented with 10% fetal bovine serum (*v*/*v*), 1% pen/strep (*v*/*v*), 1 mM sodium pyruvate and 2 mM L-glutamine. TrypLE™ Express was used to dissociate the adherent HepG2 cells from the flask prior to staining with trypan blue for cell counting and viability check using a Countess^®^ automated cell counter. Cells were seeded into Corning^®^ BioCoat™ Collagen I, 96 well, clear flat-bottom microtitre plates (Corning Life Sciences, New York, NY, USA) at a density of 1 × 10^5^ cells/mL and allowed to attach for 24 h. The cells were exposed to the test compounds prepared in the media described above for 48 h. Negative controls of 0.1% DMSO/0.4% MeOH (*v*/*v*) in media and 0.1% DMSO/media (*v*/*v*) and a positive control, valinomycin (final concentration of 60 µM), in media were prepared and included in the exposure study. Hoechst nuclear stain was used to identify cells (label DNA) and thus allowed measurement of cell number (CN), nuclear area (NA) and nuclear intensity (NI), while MitoTracker^®^ Orange CMTMRos was used to evaluate mitochondrial function such as any changes in mitochondrial mass (MM) or mitochondrial membrane potential (MMP). To study the influence of CAPP exposure on the toxicity of AMX, an untreated solution at 256 µg/mL in water (final solution in media 1.2 µg/mL) was compared with the aliquot of the same solution exposed to plasma for 20 min, resulting in 50% decomposition. The concentration chosen was similar to reported pharmaceutical serum concentrations after oral administration ranging from 0.19 to 7.61 µg/mL [48].

### 3.9. Resistance-Evolution Study and Antimicrobial Assessment of CAPP-Treated Amoxicillin

*Escherichia coli* NCTC 10418 was used as a model organism for the antimicrobial assessment of CAPP-treated AMX and for the resistance-evolution study. *E. coli* is a frequently used representative for a Gram-negative organism, a common bacterial species present in the human gut microbiome, and often the target of β-lactam antibiotics. Given its well-characterized genomic, the selection of *E. coli* offers an initial look at the potential resistance development under the treatment conditions. *E. coli* was stored at −80 °C in Microbank^TM^ vials (Pro-Lab Diagnostics, Cheshire, UK) and sub-cultured onto Müller Hinton agar (MHA) at 37 °C. Minimal Inhibitory Concentrations (MICs) were used for AMX antimicrobial assessment and were performed based on the Clinical and Laboratory Standards Institute M07-A10 “Methods for Dilution Antimicrobial Susceptibility Tests for Bacteria that grow aerobically”. For the resistance-evolution study, identical colonies of *E. coli* were used to create three bacterial suspensions in Müller Hinton Broth (MHB) (lineages A, B and C). For each lineage, bacterial suspension with either no AMX, 0.25 × MIC value of AMX (0.5 µg/mL), 0.5 µg/mL of 20 min CAPP minute-treated AMX and the equivalent AMX concentration of 20 min plasma-treated AMX (20 min CAPP treatment of a 256 µg/mL, 1 mL AMX solution in water resulted in an 80% reduction in AMX) were prepared and incubated for 16 h at 37 °C. New bacterial suspensions for each condition were propagated daily for 13 days from overnight suspension for each condition. Following overnight growth, the optical density (OD_550_ nm) of each suspension was checked to ensure adequate growth of all bacterial suspensions. Inoculation of new suspensions from the previous day’s culture was normalized to an OD_550_ nm 0.15 and diluted 1 in 200 into fresh MHB for each condition. MIC assessment was completed every two days and on the 13th day. Daily subculturing on MHA was performed on each overnight suspension to confirm the absence of contamination.

### 3.10. Computational Chemistry

Computational modeling was performed using ChemDraw Professional 18.0 Suite. A modified version of Allinger’s MM2 force field parameters served to minimize the energy and then reach target room temperature to account for possible conformational transitions (300 K, heating rate 1.0 kcal/atom/ps within 10,000 steps).

## 4. Conclusions

Results presented herein demonstrate the potential of applying CAPP as a novel antibiotic residue-mitigation strategy. High removal efficiencies were achieved in surface water, with no toxicity or contribution to the evolution of bacterial resistance detected in CAPP-treated antibiotic solutions in vitro. The results also underline the high sensitivity of CAPP-treatment efficiency to residue concentration, pH and the presence of polar macromolecules. Further research is required to fully model the CAPP-induced decontamination process taking into consideration all factors mentioned. Accordingly, implementation of CAPP may prove challenging due to the need for real-time monitoring of the influent composition as well as operational flexibility of the plasma system employed but could provide a scalable and efficient process for removal of antibiotic residues from wastewater.

## Figures and Tables

**Figure 1 antibiotics-12-01115-f001:**
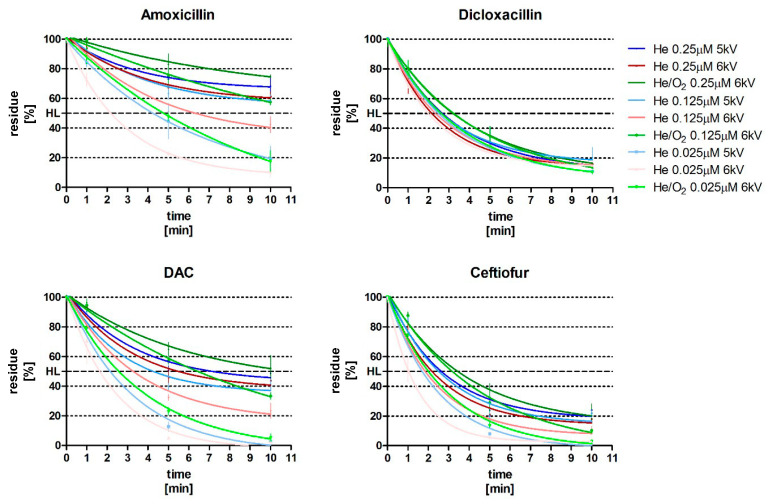
Normalized decomposition curves (mean ± SD, n = 3) representing the efficiencies of CAPP treatment. The half life (HL) is also illustrated for each compound.

**Figure 2 antibiotics-12-01115-f002:**
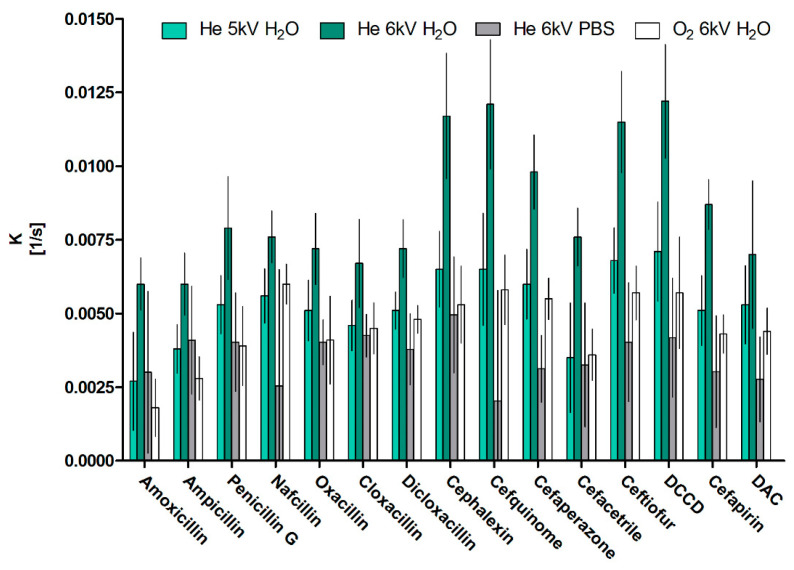
First-order decomposition rates of antibiotics (mean ± SD, n = 3) under distinct helium-based CAPP treatments: helium jet at 5 kV and 6 kV in water, helium jet at 6 kV in PBS and an oxygen-enhanced jet at 6 kV in water.

**Figure 3 antibiotics-12-01115-f003:**
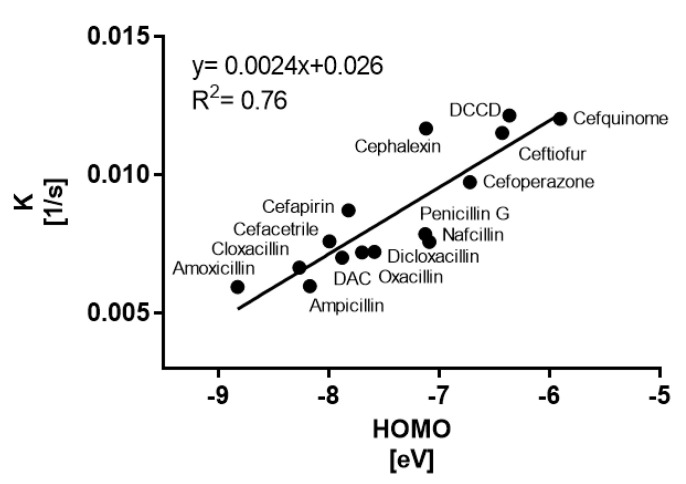
Linear regression representing the correlation between HOMO energies obtained during molecular modeling and experimental rates of CAPP (6 kV, He)-induced decomposition (k) at 25 nM in water.

**Figure 4 antibiotics-12-01115-f004:**
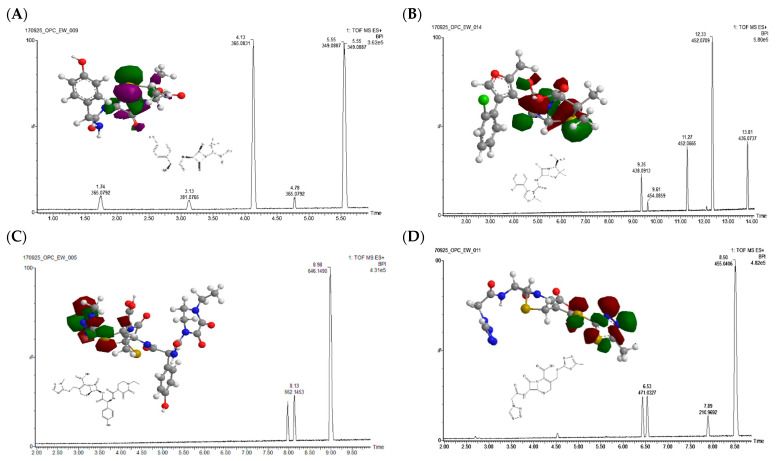
The 3D chemical structures of exemplary penicillins (amoxicillin (**A**) and cloxacillin (**B**)) and cephalosporins (cefoperazone (**C**) and cefazolin (**D**)) are shown. Each atom in the structures is represented by balls of different colors (grey for carbon, white for hydrogen, red for oxygen, blue for nitrogen and yellow for sulphur), and the highest occupied molecular orbitals (HOMOs) are overlaid on the structures in purple (for positive regions) and green (for negative regions). These are overlaid on the respective UPLC-HRMS chromatograms of CAPP-treated standard solutions at 10 µg/mL in water.

**Figure 5 antibiotics-12-01115-f005:**
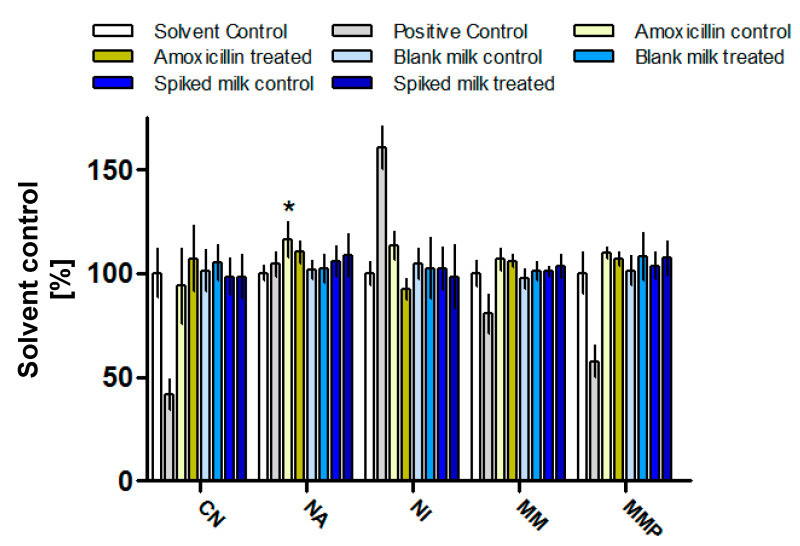
Results of 48 h exposure of human hepatocarcinoma cells (HepG2) to AMX (1.3 µg/mL), blank and spiked milk extracts treated with CAPP and respective non-treated controls (mean ± SD). Data are expressed as a percentage of untreated control ± SD. CN—cell number; NA—nuclear area; NI—nuclear intensity; MM—mitochondrial mass; MMP—mitochondrial membrane potential; responses significantly different than solvent control are denoted as * *p* ≤ 0.05.

**Figure 6 antibiotics-12-01115-f006:**
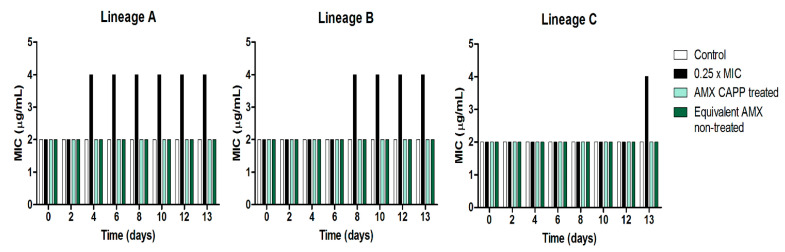
MICs of amoxicillin (AMX) from the resistance-evolution study of *E. coli* grown with sub-MIC (0.25 × MIC value) concentrations of AMX for 13 days. Lineages A, B and C are individual biological replicates taken from one colony of *E. coli* NCTC 10418.

**Figure 7 antibiotics-12-01115-f007:**
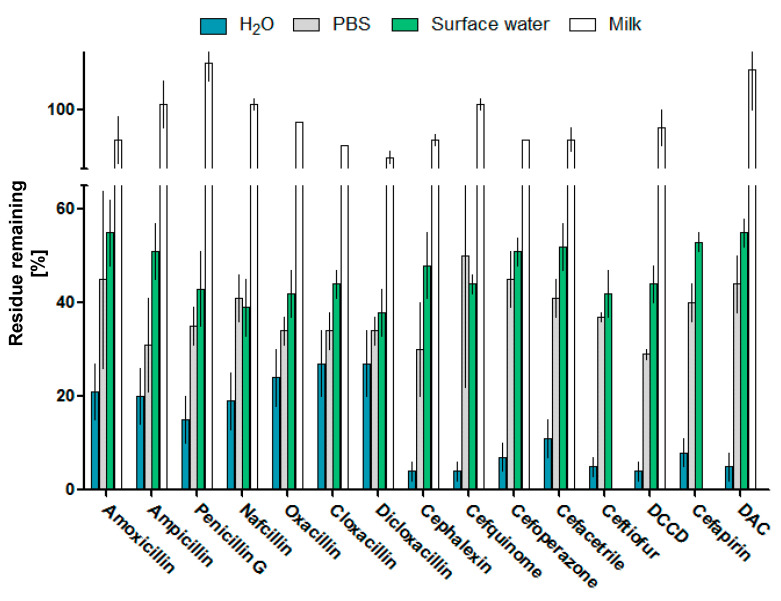
Decomposition efficiency of CAPP treatment (mean ± SD) after 5 min of exposure in water, PBS, surface water at 25 nM and milk at three times MRL.

**Figure 8 antibiotics-12-01115-f008:**
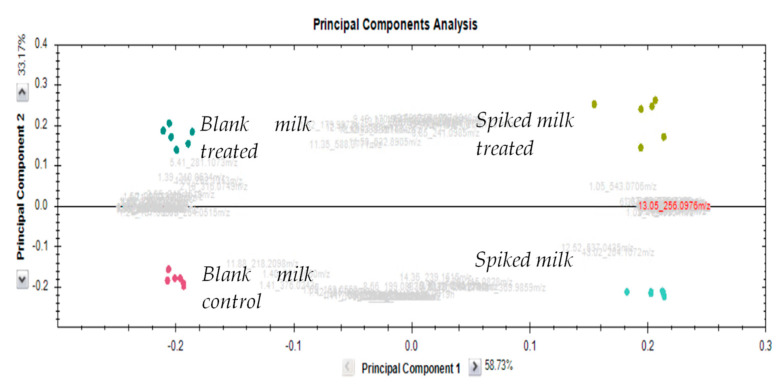
PCA score plot of significantly different UPLC-QToF features (n = 122, power 0.9995, 10% of all detected) after plasma treatment of blank and spiked milk samples. R2 (PC1-3) = 94%.

**Table 1 antibiotics-12-01115-t001:** Proposed formulae of the CAPP-derived by-products of treated antibiotics. DBE—double bond equivalent.

Compound	Retention Time	Experimental m/z	Theoretical m/z	Mass Error [ppm]	DBE	Formula	Modification
Amoxicillin	5.54	366.1113	366.1118	−1.4	9	C_16_H_19_N_3_O_5_S	
	1.74/4.13/4.78	382.1070	382.1067	0.7	10	C_16_H_19_N_3_O_6_S	+O
	3.13	398.1019	398.1016	0.6	11	C_16_H_19_N_3_O_7_S	+2O
Cloxacillin	13.81	436.0737	436.0728	2	12	C_19_H_18_ClN_3_O_5_S	
	11.28/12.33	452.0709	452.0678	6.9	13	C_19_H_18_ClN_3_O_6_S	+O
	9.62	454.0859	454.0834	5.5	11	C_19_H_20_ClN_3_O_6_S	+H_2_O
	9.36	438.0913	438.0885	6.4	12	C_19_H_20_ClN_3_O_5_S	+2H
Cefazolin	8.5	455.0406	455.0373	7.3	12	C_14_H_14_N_8_O_4_S_3_	
	7.89	210.9692	210.9664	13.2	3	C_4_H_6_N_2_O_2_S_3_	
	6.41/6.53	471.0327	471.0322	1.1	13	C_14_H_14_N_8_O_5_S_3_	+O
Cefaperazone	8.98	646.1490	646.1497	−1	17	C_25_H_27_N_9_O_8_S_2_	
	7.97/8.13	662.1453	662.1446	1.1	18	C_25_H_27_N_9_O9S_2_	+O

## Data Availability

Data is available from the authors.

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
