# Peer review of "Assessment of Cold Atmospheric Pressure Plasma (CAPP) Treatment for Degradation of Antibiotic Residues in Water"

_antibiotics, 2023, doi:10.3390/antibiotics12071115_

Round 1

Reviewer 1 Report

In the manuscript, the authors presented an study about a holistic assessment of cold atmospheric pressure plasma technology (CAPP) for β-lactam antibiotic residues removal. I commend the authors for their work and I recommend the manuscript published with the following minor revisions. Please check:

1. What do the different colorful balls mean in Figure 4?

2. How does pH affect the experimental results?

3. English should be polished.

In the manuscript, the authors presented an study about a holistic assessment of cold atmospheric pressure plasma technology (CAPP) for β-lactam antibiotic residues removal. I commend the authors for their work and I recommend the manuscript published with the following minor revisions. Please check:

1. What do the different colorful balls mean in Figure 4?

2. How does pH affect the experimental results?

3. English should be polished.

Reviewer 2 Report

The paper entitled “Assessment of Cold Atmospheric Pressure Plasma (CAPP) treatment for degradation of antibiotic residues in water” proposes a new antibiotic residue degradation/removal strategy that has been shown to be dependent on residue concentration, pH and the presence of polar macromolecules.

The paper is very well structured and written, easy to understand, providing important data regarding CAPP application.

The authors should take into consideration the comments below:

The authors evaluated whether E. coli exposed to CAPP-degraded β-lactams develop resistance. Please, specify why you chose E. coli, and why not other microorganisms besides E. coli.

Use “wastewater” or “waste water” in one form only through entire manuscript.

The title of Figure 1 is too large. Both in the text and in the picture, the working conditions are described very well, so the title of Figure 1 should be kept short. For each picture in Figure 1, place % in brackets. Please, explain HL term from Figure 1.

For all figures when “%” was used, place % in brackets.

The title and the content of Figure 2 need to be changed: "....for different settings, i.e. jet gas (helium (He) vs. helium/oxygen (O2)..." - I don't see in Figure 2 jet gas compositions as helium/oxygen.

References must be described in accordance with the Instructions for Authors. In this References list, different styles are used.

This manuscript could be accepted after a minor revision.
